# Royal Jelly: Beneficial Properties and Synergistic Effects with Chemotherapeutic Drugs with Particular Emphasis in Anticancer Strategies

**DOI:** 10.3390/nu14194166

**Published:** 2022-10-07

**Authors:** Suzy Salama, Qiyang Shou, Aida A. Abd El-Wahed, Nizar Elias, Jianbo Xiao, Ahmed Swillam, Muhammad Umair, Zhiming Guo, Maria Daglia, Kai Wang, Shaden A. M. Khalifa, Hesham R. El-Seedi

**Affiliations:** 1Indigenous Knowledge and Heritage Center, Ghibaish College of Science and Technology, Ghibaish 51111, Sudan; 2Second Clinical Medical College, Zhejiang Chinese Medical University, Hangzhou 310053, China; 3Department of Bee Research, Plant Protection Research Institute, Agricultural Research Centre, Giza 12627, Egypt; 4Faculty of Medicine, University of Kalamoon, Dayr Atiyah P.O. Box 222, Syria; 5Nutrition and Bromatology Group, Department of Analytical and Food Chemistry, Faculty of Sciences, Universidade de Vigo, 32004 Ourense, Spain; 6Faculty of Pharmacy, Menoufia University, Shebin El-Koom 32512, Egypt; 7Department of Food Science and Technology, College of Chemistry and Engineering, Shenzhen University, Shenzhen 518060, China; 8School of Food and Biological Engineering, Jiangsu University, Zhenjiang 212013, China; 9Department of Pharmacy, University of Napoli Federico II, Via D. Montesano 49, 80131 Naples, Italy; 10International Research Center for Food Nutrition and Safety, Jiangsu University, Zhenjiang 212013, China; 11Institute of Apicultural Research, Chinese Academy of Agricultural Sciences, Beijing 100093, China; 12Department of Molecular Biosciences, The Wenner-Gren Institute, Stockholm University, S-106 91 Stockholm, Sweden; 13Pharmacognosy Group, Department of Pharmaceutical Biosciences, Uppsala University, Biomedical Centre, Box 591, SE 751 24 Uppsala, Sweden; 14Department of Chemistry, Faculty of Science, Menoufia University, Shebin El-Koom 32512, Egypt; 15International Joint Research Laboratory of Intelligent Agriculture and Agri-Products Processing, Jiangsu Education Department, Jiangsu University, Nanjing 210024, China

**Keywords:** royal jelly, cancer, anticancer drugs, synergistic effect

## Abstract

Cancer is one of the major causes of death globally. Currently, various methods are used to treat cancer, including radiotherapy, surgery, and chemotherapy, all of which have serious adverse effects. A healthy lifestyle, especially a nutritional diet, plays a critical role in the treatment and prevention of many disorders, including cancer. The above notion, plus the trend in going back to nature, encourages consumers and the food industry to invest more in food products and to find potential candidates that can maintain human health. One of these agents, and a very notable food agent, is royal jelly (RJ), known to be produced by the hypopharyngeal and mandibular salivary glands of young nurse honeybees. RJ contains bioactive substances, such as carbohydrates, protein, lipids, peptides, mineral salts and polyphenols which contribute to the appreciated biological and pharmacological activities. Antioxidant, anticancer, anti-inflammatory, antidiabetic, and antibacterial impacts are among the well-recognized benefits. The combination of RJ or its constituents with anticancer drugs has synergistic effects on cancer disorders, enhancing the drug’s effectiveness or reducing its side effects. The purpose of the present review is to emphasize the possible interactions between chemotherapy and RJ, or its components, in treating cancer illnesses.

## 1. Introduction

Royal jelly (RJ) is one of the honey bee products that has attracted much attention recently [1]. RJ is a hypopharyngeal and mandibular glandular secretion of nurse honey bees. It is known as a “superfood” since it serves as the primary source of nutrition for young worker larvae for the first three days and for the lifetime of the queens [2]. RJ is produced by both the mandibular and hypopharyngeal glands, although only the hypopharyngeal gland is involved in protein synthesis [3,4]. The mandibular gland is a pair of sacs that can only be found in worker bees and the queen. On the other hand, the hypopharyngeal gland is a pair of long tuberous organs found in the worker bee’s frontal head regions. RJ is partially water soluble and highly acidic, with a gelatinous consistency. It contains proteins, water, and sugar as major components, while the minor components are composed of minerals, amino acids, vitamins, enzymes, and hormones [5]. Naturally occurring bioactive peptides comprise the highest percentage of RJ active ingredients. Bioactive peptides are responsible for most of the unique therapeutic actions of RJ [6]. For generations, fresh RJ has been used as an alternative remedy and an all-natural energy booster. It has been utilized in conventional medicine, particularly in Asia and Ancient Egypt. However, in the last years, the interest in the food industry and natural products has greatly increased, highlighting the importance of RJ and its unique pharmacological and therapeutic properties [3]. RJ has recently shown anticancer, antioxidant, anti-microbial, anti-aging, and anti-inflammatory pharmacological properties, making it a dietary supplement with functional health-enhancing criteria. It is also used in pharmaceutical preparations and as an industrial cosmetic agent [4,7].

These unique properties of RJ, besides possessing minimal toxicity compared with other bee products, make it the best choice to be combined with anticancer drugs. Cancer is a disease which has spread globally and is considered to be one of the most common causes of death worldwide. It is caused by an abnormal cell DNA mutation leading to cellular imbalance, and an uncontrolled spread prior to secondary damage of the body organs [8]. According to the World health organization (WHO) report 2050, 27 million cancer cases and 17.1 million death cases will be registered per year [9]. For the coming generations, chemotherapy complications and multidrug resistance will be a serious threat [10]. The response rates to the available cancer chemotherapeutics are decreasing. This decrease will greatly affect the present treatment protocols with time [11]. For these reasons, a need for alternative synergistic therapies or dietary supplements to face cancer becomes a must. Recent studies have greatly supported the anticancer effect of RJ, whether alone or in combination with other classical anticancer drugs. It can be used alone as an anti-proliferative entity which helps to suppress the further division of cancer cells. Similarly, it shows a synergistic interaction with anticancer drugs when used in combinations. These interactions between RJ and anticancer medications may significantly boost the drug’s therapeutic efficacy and improve the pharmacodynamic parameters. Additionally, it can demonstrate protective properties against toxicities brought on by synthesized anticancer agents [7].

The synergistic effect of RJ has several mechanisms of action depending on the anticancer drug used. Certain RJ mechanisms of action have been observed with commercially available drugs such as cyclophosphamide, 5-fluorouracil (5-FU), thymoquinone, cisplatin, temozolomide, Interferon alpha (HulFN-αN3), and GE132 plus (a nutraceutical supplement). RJ can reduce the genotoxicity and DNA damage when used with cyclophosphamide, or even decrease the cell viability and IC_50_ with 5-fluorouracil [12,13], and it can also enhance the entrance of the cells into the apoptotic cycle with Thymoquinone or cisplatin [14,15]. The synergistic effect extends to decrease cell proliferation when combined with GE132, HulFN-αN3, and Temozolomide, increasing the efficacy of these drugs [16,17,18].

One of the main challenges worldwide is to find a cure for cancer with minimal adverse effects. Traditional systemic chemo- and radiotherapies can cause severe harmful side effects, including bone marrow toxicity, hemorrhagic cystitis, gastrointestinal side effects, nausea, vomiting, diarrhea, and hair loss [19]. These side effects can be as damaging as cancer itself and can make the life of patients unbearable. Furthermore, adding other drugs to ameliorate the side effects of the anticancer drugs may trigger other health hazards. They can cause polypharmacy with expected and unexpected drug-drug interactions, exposing the patient’s health to higher risks [16]. To minimize the unwanted side effects, natural products were introduced which exhibited a synergetic effect that can diminish the undesirable side effects, improve the efficacy of the drugs, and maintain the quality of life for cancer patients [14].

In this review, we will focus on the synergistic interactions between RJ and commonly used cancer drugs as part of our ongoing projects about honeybees and bee products [20,21,22,23,24,25,26,27,28,29,30].

Authors should discuss the results and how they can be interpreted from the perspective of previous studies and of the working hypotheses. The findings and their implications should be discussed in the broadest context possible. Future research directions may also be highlighted.

## 2. Methods of Search

A literature search was conducted up until July 2022, using databases including, Sci-finder, PubMed, Scopus, Google Scholar, Science Direct and Web of Science. The search terms were addressed using the following keywords: “Royal jelly”, “anticancer drugs”,” and “Synergistic”. The search included original research articles and only articles written in the English language were considered. Research articles that investigated activities of RJ or their ingredients solely without the co-administration of pharmaceutical drugs were excluded. 72 related articles were identified via database searching; data items included main results, and interaction pathways, and the methods used to determine interaction were briefed and comprehensively described for each of the collected articles.

## 3. Anticancer Effects of Royal Jelly

It is well-known that some bee products have a positive impact on stopping cancer formation [31]. The antiproliferative activity of RJ was investigated, and RJ revealed potential anticancer properties owing to the inhibition of tumorigenesis, cancer cell proliferation and/or metastasis, via the inhibition of tumor-induced angiogenesis and/or the activation of immune function [32].

For instance, RJ exhibited a significant reduction in tumor mass and the serum concentrations of interleukin (IL)-4 and IL10, and cytokines released from type 2 T helper (Th2) cells, whereas the concentrations of IL-2 cytokines produced from type 1 T helper (Th1) cells, interferon (IFN)-α, superoxide dismutase (SOD), as well as total antioxidant machinery showed a significant elevation in an induced breast tumor. IFN-α level elevation manifested the role which RJ plays in immunomodulation. The enhanced release of antioxidant markers in the liver and kidney was another strong indicator of the antioxidant and immunomodulatory activities of RJ, with their possible association in the suppression of tumor growth in animal models [33].

However, it is also reported that RJ protein impedes the growth of human breast cancer cell lines induced by bisphenol A (BPA)—through the inhibition of the signaling pathway for cell proliferation induced by E2, and not as believed due to the inhibition of E2 attached to ER^5^. BPA is an estrogen-like substance used in commercial products such as polycarbonate plastics and synthetic resins, which are applied to coat the inside of beverage cans, and used to manufacture food wrapping paper and dental sealants. BPA has estrogenic activity and binds to estrogen receptors, stimulating the proliferation of human breast cancer MCF-7 cells [34] through the activation of the protein kinase/phosphoinositide3 kinase signaling pathway which is induced by estrogen receptor/human epidermal growth factor receptor 2 (HER2)/mitogen [35]. The crude RJ stops the damage of bisphenol A, which is a predisposition factor and hazardous insult that induces human breast cancer cell growth [34].

Recently, it was also shown that inflammation plays a significant role in the different steps of cancer cell development [36]. Equally interesting, tumor necrosis factor (TNF)-α, an inflammatory mediator, promotes cancer cell proliferation and malignant transformation by regulating other molecules. In a renal cell carcinoma (RCC) model, the level of TNF-a was surprisingly correlated with the incidence risk of the cancer, the mode of growth, invasion, and metastasis. RJ supplementation decreased the concentrations of TNF-α and transforming growth factor (TGF) -β, thus dramatically reducing the paraneoplastic syndrome in RCC patients. The observed suppression of the malignant invasiveness and the decrease in the tumor weight may be attributed to the presence of 10-hydroxy-2-decenoic acid (10-HDA), which is a remarkable fatty acid among the lipid content of RJ [37]. In comparison to other bee products, this fatty acid is exclusively found in RJ at relatively high concentrations, as illustrated in Figure 1 [38,39].

An RJ product, fraction RJP30 (RJ protein fraction), was destructive for HeLa human cervicouterine carcinoma cells. An antiproliferative effect of RJ against Lewis lung carcinoma and colorectal adenocarcinoma cells [10,15] could be related to the effect of 10-HDA, which in previous studies, constrained the initiation of transplantable leukemia in mice AKR and different ascitic tumor cell lines in mice, and showed a neurotrophic effect on murine neuronal cells [40,41]. It has been noted that 10-HDA has a similar effect to HADACIs (histone deacetylase inhibitors), a strong anti-inflammatory drug, promoting the downregulation of phosphatidylinositol 3 kinase (PI3K) and the phosphorylation of protein kinase B (AKT) [42]. It has been known that some human cancers can be induced by arylamine carcinogens. Among the known metabolic pathways for arylamine carcinogen activation is *N*-acetylation, which is believed to play a significant role in arylamine metabolism [43]. RJ interferes with the activation of arylamine carcinogens by *N*-acetylation in the presence of *N*-acetyltransferase (NAT) and acetyl CoA, precluding the metabolism of 2-aminofluorene (2-AF) metabolites in the human liver tumor cell line, and lowering the 2-AF in J5 cells [43].

The release of the pro-inflammatory cytokines TNF-α, IL-1β and IL-8 is inhibited by 10-HDA in an in vitro model, explaining its evident effect in inhibiting tumor growth and probably inducing the anti-proliferative impact seen in other types of cancers [44].

IL-6 production and nuclear factor kappa (NF-kB) activation induced by lipopolysaccharide (LPS) through either MyD88 or Toll/IL-1 receptor domain-containing adaptor-inducing IFN-b (TRIF) upregulation in murine macrophage cell line RAW264 cells, were inhibited by 10-HAD. In addition, IkB-ζ expression, and IkB-ζ-related gene production induced by LPS were specifically inhibited by 10-HDA [45]. It has been shown that the expression of extracellular SOD released by leukemia THP-1 cells is promoted by the inhibition of histone deacetylase activity by both 10-HDA and 4-hydroperoxy-2-decenoic acid ethyl ester promotion [46]. It has been reported that the stimulation of human monocyte-derived dendritic cell maturation is catalyzed by another hydroxyl fatty acid, 3,10-dihydroxydecanoic acid, which enhances their Th1 polarizing activity, suggesting a reinforcement of anticancer properties [47].

The 10-HDA and human α-interferon- (HuIFN-aN3) proteins have a similar antitumor response, and their combination decreases the level of glutathione and enhances the level of lipid peroxidation via malondialdehyde (MDA) in CaCo-2 cells [16]. RJ lipophilic fractions have an obvious anticancer effect owing to the growth regress of human neuroblastoma [48]. The rich chemical profile of RJ attributed to its effectiveness against cancer cell growth, proliferation, differentiation and migration [49].

The anticancer properties of RJ are attributed to several types of phytochemicals among which are flavonoids, such as hesperetin, naringenin, isosakuranetin, chrysin, acacetin, luteolin, apigenin, and glycoside, kaempferol and isorhamnetin glycosides, coumestrol, genistein and formononetin [50,51]. RJ also contains 57-kDa protein which stimulated hepatocyte gene expression and prolongs the proliferation of hepatocytes, as well as enhancing albumin production [52]. Therefore, such a protein has a cytoprotective effect on hepatocytes and promotes liver regeneration [53]. Major royal jelly protein 1 (MRJP 1) oligomer is a heterocomplex composed of five MRJP 1 monomers joined to one Apisimin by noncovalent bonds. It has been demonstrated that MRJP 1 promoted cell proliferation in a human T-leukemia derived lymphoid cell line [54].

## 4. Interaction with Anticancer Drugs

Of the most deadly and serious illnesses is cancer, since it causes a high morbidity and mortality worldwide. Based on the global cancer records (GLOBOCAN) published in 2020, it was approximated that more than 19 million new cancer patients are to be reported from 185 countries and with 36 cancer-types [55]. Equally worrisome is the predicted record of an average 33 million new cases by the year 2040, as stated by the World Health Organization [14,56]. As per the complicity of cancer formation and progression, multi-composite synergistic pharmacological solutions and drug reorientation were suggested by Nogales et al., 2021 [57]. Among them is the potential synergism of one or more hive products with other commercially available anticancer drugs, either to strengthen the efficacy of the drug or to eliminate the side effects of the drug [12,58]. RJ has received widespread recognition as a safe agent in previous trials. Briefly, oral treatment of 10 g/kg RJ in a mouse model demonstrated no acute toxicity [59], and it was speculated that consuming RJ may alter the function of the spleen and thymus through cell-mediated and humoral immunity [33]. Still, RJ carries a great potential for health empowerment when administrated in connection to anticancer drugs, as reviewed in the following sections.

### 4.1. Thymoquinone

Thymoquinone is a bioactive component of the *Nigella sativa* plant and has been proved effective in treating cancer, explained by its immunoenhancing and anti-oxidative properties [60]. Moubarak et al. demonstrated that the administration of RJ at a dose of 5 µg/mL combined with thymoquinone at the concentration 10 µmol/L for 24 h could reveal the remarkable induction of the caspase-3 apoptotic pathway and the death of human breast cancer cells MDA-MB-231. These findings were reproduced in different cell lines with similar effects and without any cytotoxic changes to normal human small intestinal cells [14].

### 4.2. Temozolomide

Temozolomide is a conventional chemotherapy used against brain cancer, and approved by the Food and Drug Administration (FDA) in 2005 [61]. The addition of RJ extract (30 µg/mL) to temozolomide (20 µM) has synergistically elevated the cytotoxicity of the drug on the human glioma cell line U87MG [7].

### 4.3. Interferon Alpha

Interferon alpha is a multi-subtype protein which is used as an antitumor drug. RJ and its active compound 10-HDAA were found to improve the potency of HulFN-αN3 when applied to the human colorectal adenocarcinoma cell line CaCo-2. The same report claimed the mechanism of action to be due to the augmentation of lipid peroxidation, the inhibition of cell proliferation and the depletion of glutathione (GSH) levels in colon cancer cells [16].

### 4.4. GE132 Plus

GE132 plus is a nutraceutical supplement which contains five strong antioxidant compounds, *Ganoderma lucidum* extract, RJ, resveratrol, sulphraphane and lycopene in one capsule (500 mg) [17]. In Traditional Chinese Medicine, it has been utilized for decades as a health-promoting factor for treating various diseases owing to its therapeutic properties, including the anticancer, immunopotentiator, and anti-hypertensive impacts [62]. Significant anti-proliferative effects through the potential inhibition of angiogenesis have been recognized upon administration of GE132 plus to breast cancer cell lines (MCF-7), prostate cancer cell line (PC3), and the human colorectal adenocarcinoma cell line (SW48). On the other hand, GE132 plus did not show significant cytotoxicity on mesenchymal stem cells or peripheral blood collected from healthy donors in vitro. The same report revealed that higher concentrations of GE132 plus (500–2000 µg/mL) exhibited a cytotoxic effect to normal human vascular endothelial cell line EA.hy 926. That cytotoxic effect was attributed to the possible cancer-like mechanism of one of the drug’s components on the hybrid cell line EA.hy 926 which was immortalized by the integration between normal cells and adenocarciomatous cells [17].

### 4.5. Cisplatin

Although modern chemotherapeutic drugs have shown an efficient improvement in patient survival, the adverse effects of these drugs represent a great concern for cancer patients as well as physicians. Cisplatin is a well-known drug that is used in many diseases [63] and an active drug in anticancer chemotherapy as well, yet its adverse hepatotoxic and nephrotoxic effects are the biggest challenges facing cancer patients [64]. Some in vivo and clinical studies have shown that the synergistic interaction of RJ with cisplatin, when both are applied simultaneously, has considerably attenuated the previously observed nephrotoxic and hepatotoxic effects. Ali et al. 2011, reported that the daily oral gavage of RJ (300 mg/kg) followed by one of cisplatin intra-peritoneal doses (7 mg/kg) to Sprague Dawley rats could notably reduce apoptotic damage and lipid peroxidation, and elevate the endogenous antioxidant enzymes [15]. Further, a sub-chronic toxicity study was performed on Wister rats, where the rats were treated with an oral gavage dose of RJ (100 mg/kg) one hour before intra-peritoneal injection of cisplatin (1 mg/kg) for two weeks [65]. The findings suggested that the production of the fibrogenic factorsTGF-β1 and alpha smooth muscle actin (α-SMA) in the kidney tissues was significantly reduced. The synergistic protective effect of RJ with cisplatin has been confirmed by a clinical study. The intake of RJ pre- and post-cisplatin chemotherapy administration showed obvious protection to kidney tissue, as the serum level of creatinine and urea was stable both pre and post the cisplatin regimen [66]. RJ at a dose of 5 g/day was administrated for 30 days and proven effective in preventing the development of oral mucositis following a planned radiotherapy and cisplatin chemotherapy in head and neck cancer patients involved in a small population-sized clinical trial [67]. In another similar clinical trial, cancer patients receiving a variety of anticancer therapies were supplemented with 5 mL/day of processed honey mixed with RJ for 30 days. The main finding was the relief from the cancer-related fatigue symptoms using this combination compared with RJ alone [68].

### 4.6. Cyclophosphamide

Cyclophosphamide is another neoplastic drug used in treating breast cancer patients. Its mechanism relies on the generation of hepatic cytochrome P_450_ enzymes in liver cells [69]. These cytochrome enzymes play a crucial role in the metabolic pathways of anticancer drugs in the liver, and are accordingly responsible for their cytotoxicity in the patient’s body [70]. Earlier reports supported the notion that the cyclophosphamide regimen reflects adverse effects through the damage to the reserve oocytes’ nuclei in female patients [71]. Fahmy et al. 2015, concluded that the synergistic effect of honeybee products, including RJ, with cyclophosphamide, can counteract the adverse effects of introduced cyclophosphamide. A significant reduction in the genotoxicity was observed after intake for 15 days, where the DNA damage to liver cells of treated mice was attenuated [12].

### 4.7. 5-Fluorouracil

The anticancer drug 5-FU is considered one of the most recommended drugs in treating patients with colorectal cancer through the suppression of the thymydilate synthase enzyme, the key player in the colon cancer cell cycle [72]. The incubation of the human colon cancer cell line HTC-116 with a mixture of wild RJ and 5-fluorouracil induced a remarkable decline in the cell viability, along with a 45% decrease in the value of IC_50_ in comparison with 5-FU alone [13].

The effect of the synergistic interaction of RJ with the main chemotherapeutic anticancer drugs is summarized in Figure 2 and Table 1, and represents the increase in the apoptosis of cancer cells, as well as the reduced cytotoxicity in normal cells.

## 5. Potential Limitations

Although our search strategy highlighted many in vitro and preclinical in vivo positive findings which support the synergism between RJ and anticancer drugs, some limitations have been raised. Negative results have been revealed from some synergistic anticancer activity studies, especially when testing high doses of the GE132 plus drug in vitro, as mentioned in the present review. In addition, a few clinical confirmatory studies have been conducted on the topic. Moreover, not all the studies showed a clear and detailed mechanistic explanation as to the action of RJ in reducing the side effects of anticancer drugs.

## 6. Conclusions

Collectively, the present review gives detailed and updated data for the anticancer activity of RJ. The collected data revealed significant protection properties of RJ against different types of cancer, which was attributed to its active compounds. Additionally, this review exposed the synergistic interaction of RJ with commonly used cancer chemotherapy, either through its inhibitory effect against the adverse effects of the drugs, or through its enhancement of the anticancer-potential of the drug. Moreover, the current review can provide researchers with the data required to conduct further studies on the anticancer potential of the compounds purely isolated from RJ.

## Figures and Tables

**Figure 1 nutrients-14-04166-f001:**
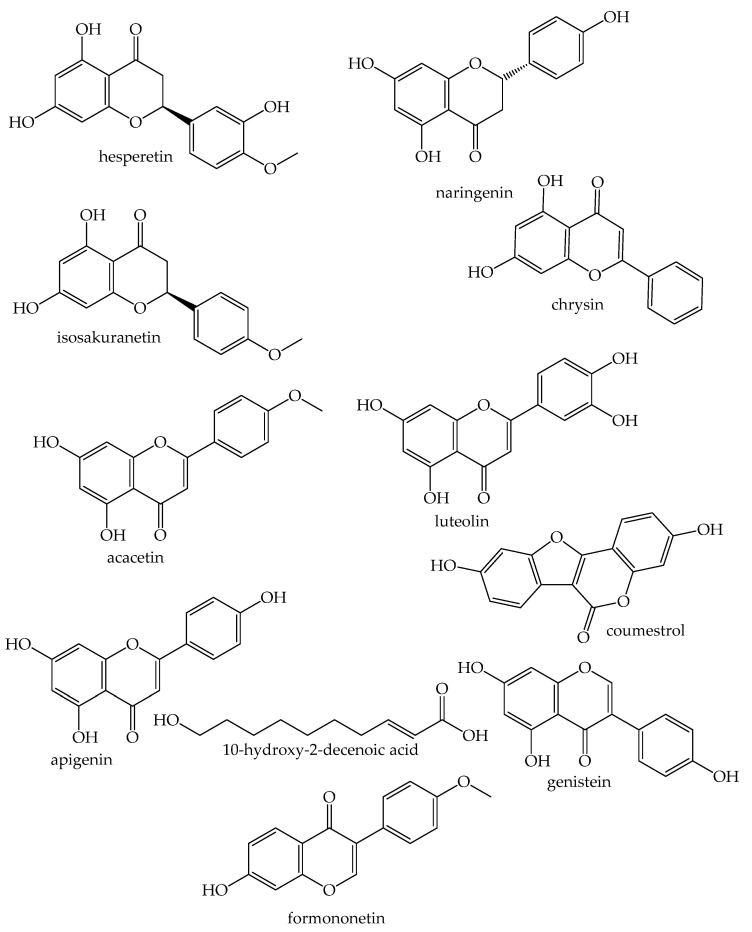
Identified anticancer compounds from royal jelly.

**Figure 2 nutrients-14-04166-f002:**
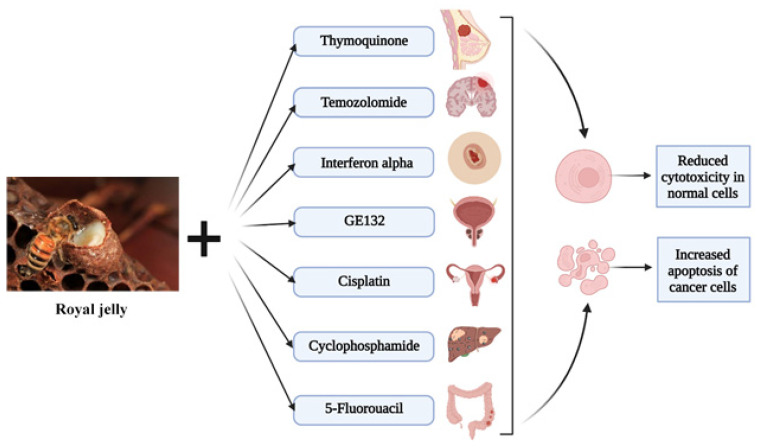
The effect of the synergistic interaction of royal jelly with anticancer drugs on cancer cells as well as normal cells.

**Table 1 nutrients-14-04166-t001:** Synergistic interaction between royal jelly and anticancer drugs.

Drug	Royal Jelly Sample	Experimental Subjects	Mechanism Of Synergistic Action	References
Thymoquinone	Crude extract	Human breast cancer malondialdehyde (MDA)-MB-231 and human small intestine FHs-74 cell lines	Enhancement in the apoptotic cell death of breast cancer cells via a significant reduction in cell proliferation and remarkable expression of the caspase-3 apoptotic pathway without exerting a cytotoxic effect on normal small intestine cells.	[14]
Temozolomide	Royal jelly (RJ) extract	Human glioma cell line U87MG	Increased glioma cell line cytotoxicity and decreased cell proliferation.	[7]
Interferon alpha (HulFN-αN3)	Fresh RJ or its active component 10-hydroxy-2-decenoic acid (10-HDAA)	Human colorectal adenocarcinoma cell line CaCo-2	Improved augmentation of lipid peroxidation, suppression of cell division and decrease in the glutathione level of colon cancer cell line when treated with RJ + HulFN-αN3. Lower effect was observed with 10-HDAA + HulFN-αN3.	[16]
GE132 plus	RJ extract	Breast cancer cell lines MCF-7, Prostate cancer cell line PC3, human colorectal adenocarcinoma cell line SW48, normal human vascular endothelial cell line EA.hy 926, dental mesenchymal stem cells and peripheral blood cells.	Significant suppression of breast, colon and prostate cancer cell proliferation. Inhibition of proliferation of vascular endothelial cells at high concentrations only, without any effect on mesenchymal stem cells or blood cells count.	[17]
Cisplatin	RJ extract RJ capsules RJ capsules Organic RJ Mixture of processed honey and RJ	Sprague Dawley rats Wister albino rats Cancer patients Head and neck cancer patients Variety of cancer patients	Anti-apoptotic activity of kidney and liver cells via the decrease in MDA and increase in glutathione (GSH), glutathione peroxidase (GSH-Px), glutathione-s-transferase (GST) and superoxide dismutase (SOD) contents of cells. Reduction in nephrotoxicity through the inhibition of serum urea, creatinine and uric acid. Significant decrease in the expression of the fibrosis factors, transforming growth factor (TGF-β1) and alpha smooth muscle actin (α-SMA). Reduced development of oral mucositis. Suppression of cancer-related fatigue in patients on different anticancer therapies	[15,65,66,67]
Cyclophosphamide	Combination of honey + RJ + pollen grains	Mice	Reduction in cyclophosphamide genotoxicity through amelioration of DNA damage.	[12]
5-Fluorouracil (5-FU)	Wild RJ	Human colon cancer cell line HCT-116	Suppression of cytotoxic effect of 5-FU through arresting cancer cell division and growth.	[13]

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
