# Peer review of "Royal Jelly: Beneficial Properties and Synergistic Effects with Chemotherapeutic Drugs with Particular Emphasis in Anticancer Strategies"

_nutrients, 2022, doi:10.3390/nu14194166_

Round 1

Reviewer 1 Report

Authors submitted a well written and comprehensive review of royal jelly and its influence on chemotherapy and related drugs in relationship to cancer mechanisms.

While the authors provide a compelling case, there does not appear to be a balanced view of royal jelly.  Almost all papers reviewed came from cell lines and animal studies.  Were there no clinical studies?  One was included in the discussion of cisplatin, but it is unclear whether there is a lack of clinical evidence.  Authors need to add discussion about potential limitations, e.g. how cell lines and animal studies may provide preliminary evidence but are not definitive.  The authors' case would be more compelling by explaining potential limitations of their review.  Were there studies that found no association between royal jelly and cancer effects?  Does royal jelly have any side effects?  

Author Response

File attach

Reviewer 2 Report

The presented manuscript is very interesting. The paper is well prepared.

In my opinion, it should be published.

Author Response

Comments and Suggestions for Authors

The presented manuscript is very interesting. The paper is well prepared.

In my opinion, it should be published.

Response: We would like to thank the reviewer for the nice words

Reviewer 3 Report

Attach file

Author Response

File attach
